# Contrasting Regulators of the Onset and End of the Seed Release Phenology of a Temperate Desert Shrub *Nitraria tangutorum*

**DOI:** 10.3390/plants12010088

**Published:** 2022-12-23

**Authors:** Fang Bao, Zhiming Xin, Minghu Liu, Jiazhu Li, Ying Gao, Qi Lu, Bo Wu

**Affiliations:** 1Institute of Desertification Studies, Chinese Academy of Forestry, Beijing 100091, China; 2Institute of Ecological Conservation and Restoration, Chinese Academy of Forestry, Beijing 100091, China; 3Key Laboratory for Desert Ecosystem and Global Change, Chinese Academy of Forestry, Beijing 100091, China; 4Gansu Minqin Desert Ecosystem National Observation Research Station, Wuwei 733300, China; 5Experimental Center of Desert Forestry, Chinese Academy of Forestry, Bayannaoer 015200, China

**Keywords:** desert species, seed release, water addition, phenology

## Abstract

Seed release is crucial in the reproductive cycle of many desert plant species, but their responses to precipitation changes are still unclear. To clarify the response patterns, we conducted a long-term in situ water addition experiment with five treatments, including natural precipitation (control) plus an extra 25%, 50%, 75%, and 100% of the local mean annual precipitation (145 mm), in a temperate desert in northwestern China. Both the onset and end of the seed release phenophase of the locally dominant shrub, *Nitraria tangutorum*, were observed from 2012 to 2018. The results showed that both the onset and end time of seed release, especially the end time, were significantly affected by water addition treatment. On average, the end time of seed release was advanced by 3.9 d, 7.3 d, 10.8 d, and 3.8 d under +25%, +50%, +75%, and +100% water addition treatments, respectively, over the seven-year study, compared with the control. The changes in the onset time were relatively small (only several hours), and the duration of seed release was shortened by 4.0 d, 7.5 d, 10.8 d, and 2.0 d under +25%, +50%, +75%, and +100% water addition treatments, respectively. The onset and end time of seed release varied greatly between the years. Preceding fruit ripening and summer temperature jointly regulated the inter-annual variation of the onset time of seed release, while the cumulative summer precipitation played a key role in driving the inter-annual variation of the end time. The annual mean temperature controlled the inter-annual variation of the seed release duration, and these interactions were all non-linear.

## 1. Introduction

Seed dispersal links the reproductive cycle of adult plants with the establishment of their offspring and is a fundamental process affecting the destinies of seeds and seedlings. It also determines the regeneration dynamics and distribution patterns of plant communities [1,2]. Therefore, the timing of seed dispersal is consequently critical in regulating the plant’s distribution, abundance, and population dynamics [3,4] because it determines the future locations where seeds and later seedlings will survive or die [3,5] or the activity changes of animals depending on the fruits and seeds [6]. For example, the time of seed release determines the fecundity and mortality of granivores, which respond quickly to annual changes in the seed change [7]. The granivorous ants and rodents primarily harvest recently produced seeds [8,9], and harvester ants’ diets depend on the temporal patterns and availability of fresh seeds from the preferred plant species [8].

The seed shattering phenology of weeds has attracted much attention because it determines the efficacy of harvest weed seed control (HWSC) at crop harvest [10,11,12]. The time of seed release is regulated by complex physiological and genetic mechanisms in conjunction with environmental factors [10]. Inter-annual variation in climate factors, such as temperature, precipitation, and day length, can lead to changes in the seed release phenology, number and activity of seed dispersers, seed predators, and competitors [1,13,14,15]. There is a general consensus that water relations play a critical role in seed development and maturity [16,17,18,19]. For example, the timing of seed release of the desert annual *Belpharis sindica* is called rain-cued seed release because it is driven by rainfall in the Thar Desert, India [17]. Seed release correlated significantly with the rain events of the black cottonwood forest along the lower Duncan River in southeastern British Columbia, Canada [18]. Successful seed germination and seedling establishment of many desert species depend on the pattern and duration of seed rain [2,20]. However, studies on the annual differences in the seed release time of desert shrubs and their drivers have not been reported.

Nitraria, a small genus of the Nitrariaceae or Zygophyllceae family, are widely distributed in the desert and arid areas of Asia, Europe, Africa, and Australia [21,22]. Nitraria comprises seven verified species, *Nitraria pamirica*, *N. retusa*, *N. roborowskii*, *N. schoberi*, *N. sibirica*, *N. sphaerocarpa*, and *N. tangutorum* [23], of which three can be found in China [24]. *N. tangutorum*, a native perennial species, is widely distributed throughout the northwestern regions of China. It is often used for sand and soil conservation due to its resistance against wind, salt, drought, cold, and sand stresses [21,25,26,27]. The species exhibits an exceptional ability in trapping wind-driven sediments around vegetation and forming phytogenic mounds, ‘nabhkas’; thus plays a pivotal role in desert ecology by fixing sands and reducing the deserts’ expansion rate [28,29]. Furthermore, leaves of the Nitraria species are used as nutritional and healthy forage for camels and sheep [30,31]. Fruits of *N. tangutorum*, the so-called “desert cherry,” are used to prepare drinks, fruit wines, and vinegars, and as a traditional medicine to treat dyspepsia, neurasthenia, abnormal menstruation, and heart disease [32,33,34]. Several phytochemical identification and extraction analyses have also been conducted on the fruits, seeds, and leaves of Nitraria to fully use the species’ commercial value [35,36,37,38]. For instance, the phenols, alkaloid, flavonoids, and polysaccharides extracted from the seeds have been confirmed to possess antioxidant and health benefits [39,40,41,42]. Despite its ecological, economic, and medicinal values, little is known about the factors determining the onset and end of seed release time in *N. tangutorum*.

Earlier studies showed that high precipitation induces early leaf unfolding and delayed leaf coloring, and delayed leaf fall in *N. tangutorum* [43,44]. In the desert ecosystem, the regeneration of the *N. tangutorum* population relies mainly on the formation of new nebkha, which is essential for compensating for the ageing nebkhas, and their spatial expansion relies mainly on seed germination in new locations [5]. The increasing precipitation trend over the last 40 years in the present study area matches the regional trend of northwestern China [45,46,47]. To investigate the prospective coordination between environmental conditions and seed release, we observed the seed release phenology of *N. tangutorum* every two days over seven years from 2012 to 2018. Five water addition treatments were designed to simulate precipitation increase: natural precipitation (control), and natural precipitation plus an extra 25% (+25%), 50% (+50%), 75% (+75%), and 100% (+100%) of the local annual mean precipitation (145 mm, 1978–2008). Both the onset and end of the seed release phenology were detected. The aims of this study were to address the following specific questions: (1) How does water increase influence the seed release phenology of *N. tangutorum*? (2) Does water affect the variation of seed release time? (3) Do the onset and end of seed release shift consistently with each other?

## 2. Results

### 2.1. Interannual Dynamics of Meteorological Factors

Inter-annual changes in annual mean temperature (AMT) during winter (T_Win_, December to February), spring (T_Spr_, March to May), summer (T_Sum_, June to August), autumn (T_Aut_, September to November), the annual precipitation (AP), and cumulative precipitation during winter (P_Win_), spring (P_Spr_), summer (P_Sum_), and autumn (P_Aut_) have been reported in a study by Bao et al., 2020. All these climatic factors showed no significant increasing or decreasing trends during the seven-year study period. This study focused on the variations of the accumulated precipitation (Figure 1A) and the mean temperature (Figure 1B) before and during the seed release period (1 June–31 August). The accumulated precipitation during the summer of 2012, 2013, 2014, 2015, 2016, 2017, and 2018 were 178.5 mm, 43.6 mm, 38.5 mm, 42.3 mm, 160.5 mm, 47.4 mm, and 55.1 mm, respectively, which contributed to 84%, 74%, 40%, 29%, 85%, 55%, and 51% of the annual precipitation, respectively. Based on the mean deviation value of the seven years, 2012 and 2016 were “above–average” (i.e., wet) summer years, while 2013, 2014, 2015, 2017, and 2018 were “below–average” (i.e., dry) summer years (Figure 1A).

### 2.2. Changes in Seed Release

Linear mixed model analysis showed that water addition treatments, year, and the preceding phenological event (peak of the fruit ripening, PFR) had significant effects on the onset (OSR), end (ESR), and duration (DSR) of seed release (all *p* ≤ 0.05, Table 1) over the seven years (2012–2018). The PFR and year were significantly associated with OSR and ESR (all *p* = 0.02, Table 1), while water addition treatment and PFR were significantly associated with DSR (*p* < 0.01, Table 1). There was no significant interaction between the three factors and OSR, ESR, and DSR (all *p* > 0.05, Table 1).

The OSR was advanced in two (2012 and 2014) and delayed in three (2013, 2015, and 2016) of the seven years for the four water addition treatments (+25%, +50%, +75%, and +100%) (all *p* > 0.05 with exceptions of +100% water addition treatment in 2012, and +25%, +50%, +75% water addition treatments in 2016, Figure 2A). Shifting directions of OSR were consistent in five (2012–2016) but different in two years (2017, 2018) of the seven years when the amount of water added increased (Figure 2A). On average, over the seven years (2012–2018), the changes of OSR in +25%, +50%, +75%, and +100% water addition treatment plots were relatively small (less than 0.20 d) compared with the control, and there was no significant difference between the treatments (all *p* > 0.05, Figure 2A, last column).

The ESR was advanced in all seven years (2012–2018) and all four water addition treatments (Figure 2B). On average, the occurrence of ESR in +25%, +50%, +75%, and +100% water addition treatment plots was significantly advanced by 3.9 d, 7.3 d, 10.8 d, and 3.8 d, respectively, compared with the control over the seven years (2012–2018) (*p* < 0.05 under +50% and +75% treatments, Figure 2B, last column). The shifting direction of the OSR and ESR were consistent in two (2012 and 2014) but inconsistent in all the other five years of the seven years (2012–2018) (Figure 2A,B).

The DSR was shortened in five of the seven years (2012–2018) under all the four water addition treatments (all *p* > 0.05 with exceptions of +100% treatment in 2014 and all water addition treatments in 2016, Figure 2C). However, the DSR was prolonged in 2014 and only significantly under +100% water addition treatment (Figure 3B). On average, over the seven years (2012–2018), the DSR was significantly shortened by 4.0 d, 7.5 d, 10.8 d, and 2.0 d under +25%, +50%, +75%, and +100% water addition treatment, respectively, compared with the control (*p* < 0.05 under +50% and +75% treatments, Figure 2C, last column).

### 2.3. The Correlations between Seed Release Events and Water Addition Amounts

A simple linear regression analysis showed no significant correlations between OSR and water addition amounts. Similarly, no significant correlation was found between ESR and water addition amounts in all seven years except 2012 (Figure 3B). Except in 2017, there was also no significant correlation between DSR and water addition amounts in all seven years (Figure 3C).

### 2.4. The Correlations between Seed Release Events and Meteorological Factors

The seed release time varied greatly from year to year. A simple linear regression analysis showed marginally significant (control, 0.05 ≤ *p* ≤ 0.1, Figure 4A, Table 2) or significant (+25%, +50%, +75%, and +100% water addition treatment, all *p* ≤ 0.05, Figure 4B–E, Table 2) downward quadratic correlations between OSR and the summer mean temperature (T_sum_). However, there was no significant correlation between the other meteorological factors and OSR (Table 2).

Unlike OSR, ESR was not correlated to T_sum_, but it significantly correlated with cumulative summer precipitation (P_sum_) under all treatments (control, +25%, +50%, +75%, and +100%), with upward quadratic relationships (all *p* ≤ 0.05, Figure 5, Table 2). Similarly, there was no significant correlation between ESR and other meteorological factors (Table 2).

AP and AMT represent the annual precipitation and annual mean temperature, respectively. T_Win_, T_Spr_, T_Sum_, and T_Aut_ represent the mean temperature during the winter (December to February), spring (March to May), summer (June to August), and autumn (September to November), while P_Win_, P_Spr_, P_Sum_, and P_Aut_ designate the accumulated precipitation during the winter, spring, summer, and autumn, respectively. OSR, ESR, and DSR represent the onset, end, and duration of the seed release phenophase.

There were marginally significant (+50%, 0.05 ≤ *p* ≤ 0.1, Figure 6C) or significant (control, +25%, +75%, +100%, all *p* ≤ 0.05, Figure 6A,B,D,E) downward quadratic relationships between the annual mean temperature (AMT) and DSR. No significant correlation was found between the other meteorological factors and DSR (F 2).

### 2.5. The Correlations between Seed Release Events and the Other Phenological Events

The inter-annual ESR change did not correlate to any of the early-season leaf phenological events, onset or end of leaf unfolding, under the water addition treatments (control, +25%, +50%, +75%, +100%) over the seven years (2012–2018) (all *p* > 0.05). Contrarily, the inter-annual OSR changes significantly correlated with the preceding fruit ripening events, i.e., fruits developing red colors, under all treatments (control, +25%, +50%, +75%, +100%, all *p* < 0.05, Figure 7) over the seven years (2012–2018).

### 2.6. The Correlations between Seed Release Events and Seed Size and Weight

The OSR changes did not correlate with the changes in seed size (volume and surface area) and weight (Thousand Seed Mass) with increasing water amount (Figure 8A–C) in 2017. However, even though all *p*-values were greater than 0.05 (*p* > 0.05), there were good upward quadratic relationships between ESR and seed volume (Figure 8D), seed surface area (Figure 8E), and seed weight (Figure 8F). The change in DSR also had a quadratically upward association with the changes in seed volume (Figure 8G, *p* > 0.05), seed surface area (Figure 8H, *p* < 0.05), and seed weight (Figure 8I, *p* > 0.05) with the increasing water amount in 2017.

## 3. Discussion

### 3.1. Effects of Water Addition Treatments on Seed Release Events

Earlier work indicated that, unlike folia phenology, the fruiting phenology, including fruit setting and ripening, of *N. tangutorum* was not significantly affected by water addition treatments [29]. In this study, we found similar results, indicating that water addition treatment almost did not change the OSR in most cases. On average, the OSR changed only several hours after water addition treatment over the seven years (Figure 2). However, ESR was significantly advanced by water addition treatment, possibly because the *N. tangutorum* plants in the water addition treatment plots bore smaller seeds than those in control plots. Seed volume, seed surface area, and the Thousand Seed Mass declined significantly with increasing water addition [6,48]. The smaller seeds produced by maternal plants from water addition treatment plots might have been due to three potential reasons. Firstly, after a long-term water addition treatment without nitrogen addition, the soil depleted gradually [49], limiting nitrogen and constraining seed development [50]. Secondly, seed size and mass usually indicate potential seedling survivorship [51], and seedlings from larger seeds have an initial advantage over smaller seeds due to the ability to establish, especially in stressful environments [52,53]. We speculated that as the degree of nitrogen limitation increased in water addition treatment plots, the number of plants that could be contained in the nebkhas reduced. Thus, the maternal plants of *N. tangutorum* adapted to this phenomenon by producing several smaller seeds to reduce the chances of seedling competition in the same nebkha. The germination rate of seeds from the control, +25%, +50%, +75%, and +100% water addition treatment plots decreased by 43%, 40%, 38%, 35%, and 32%, respectively [48], support our speculation. The “negative density-dependent recruitment” hypothesis, which states that larger seeds are more likely to produce surviving seedlings than smaller seeds, also explains this phenomenon. The third reason for the smaller seed size is that seed germination must coincide with unpredictable rainfall in desert ecosystems where water is the most limiting factor for seeding establishment. Therefore, plants in these ecosystems produce smaller seeds because they disperse to longer distances than larger ones, enabling seedling establishment in new locations [1].

At least, in theory, earlier seed release time might have limitations, e.g., lower fecundity, because advanced maturity shortens the growth period, results in smaller seeds, and requires minimal resources for reproduction. We found close correlations between seed size, seed mass, and ESR and between seed size, seed mass, and DSR in 2017 (Figure 8). Based on these findings, we speculate that smaller seeds from water addition plots matured faster and shed earlier from maternal plants than the larger seeds from the control plots. The environmental conditions experienced by parental plants directly affect their seed and infructescence traits, and the plants will be forced to maintain the quality of their offspring at a stable level and ensure the seedlings that develop can adapt to the new environment [54]. The shortened retention time of *N. tangutorum* seeds on the maternal plant most likely serves primarily to effect temporary dispersal. Contrary to our findings, a previous study conducted in Israel reported that with increasing aridity, two annual grass species, *Hordeum spontaneum* and *Avena sterilis*, produced smaller seeds [55]. This difference may be attributed to the different research scales, including the regional and precipitation scales of the two studies. Our study focused only on one location experiencing less than 200 mm of rainfall; however, the contrasting study was conducted on a much larger region with a precipitation gradient ranging from 90 mm to 1600 mm.

### 3.2. Drivers of Inter-Annual Variations of Seed Release Phenology

Seed release timing is highly variable between years and varied inter-annually by many days in the present study. This variation might affect many aspects of the desert-like granivore-seed interactions, community ecology, and ecosystem function [8,56,57]. We hardly found seeds beneath the shrubs in the study sites during the whole seed rain period, implying that seeds of *N. tangutorum* were collected or harvested soon after being released before entering the seed bank. However, it is not yet clear how time changes in the release of *N. tangutorum* seeds will affect the granivores, thus necessitating further studies.

The finding that the drivers of the onset and end of seed release of *N. tangutorum* are different in the present study is intriguing. The inter-annual variation of OSR was significantly related to the occurrence of preceding fruiting phenology (Partial Eta Squared; 0.64) and summer air temperature (Partial Eta Squared; 0.23), with the former exhibiting a higher correlation than the latter. This indicated that the OSR is a genetically controlled trait, regulated or influenced by temperature during the seed formation period; however, upon seed release, the ESR was completely dominated by environmental factors. The accumulated precipitation during seed formation determined the end time of seed release in the present study. The fact that the OSR and ESR exhibited opposite trends after the water addition treatments in most years of the seven-year study period proves their contrasting inter-annual regulators.

## 4. Materials and Methods

### 4.1. Study Area

The study was conducted in a desert ecosystem water addition platform (106°43′ E, 40°24′ N, 1050 m a.s.l.) located in Dengkou County, Inner Mongolia, China. The average precipitation between May and September was 145 mm (1978–2008), followed by a cold and dry winter. The annual average temperature and evaporation during that period were 7.6 °C and 2381 mm, respectively. The *N. tangutorum* shrub species dominating the study site often forms phytogenic nebkhas distributed in patches on the surface of hard mud. The nebkhas are approximately 1~3 m high and 6~10 m wide, with vegetation coverage of approximately 45~75%.

### 4.2. Simulated Precipitation Enhancement

Five water addition treatments, including 0% (control), 25% (+25%), 50% (+50%), 75% (+75%), and 100% (+100%) of the local annual average precipitation (145 mm), were designed to simulate precipitation increase. The water addition treatments were applied equally every month from May to September, and the additional water amounts were 0, 7.3, 14.5, 21.8, and 29.0 mm each time for the five water addition treatments, respectively. The water was pumped into a water tank from a well near the plots with water meters and transported to each sprinkler. The sprinklers, with two automatically rotating spraying arms (6 m in length) that could uniformly sprinkle water over the treatment area, were installed on the top of each nebkha (plot). More detailed information on the experimental design and the irrigation system can be found in our previous publications [43,44].

### 4.3. Phenological Observations

The phenology data were collected from 2012 to 2018 following the standard protocols of the Phenological Observation Methodology in China [58]. Phenology observation started in March 2012 until November 2018. Phenological events, including leaf unfolding (onset and end), flowering (onset, peak, and end), fruit setting (onset, peak, and end), fruit ripening (onset, peak, and end), seed release (onset and end), leaf coloring (onset and end), and leaf fall (onset and end), were recorded every two days for all shrubs in each nebkha (plot). The present study mainly focused on the seed release phenology; the detailed observation methods of each phenological event have been reported in our earlier publication [29]. Days at which at least one or several red berries fell off from the whole nebkha were recorded as the onset of seed release, while days with more than 90% of the red berries falling off from the whole nebkha were recorded as the end time of seed release. Precipitation, air temperature, relative humidity, and evaporation data were continuously recorded by a standard meteorological station near the experimental plots. The oven-drying method was used to measure the soil’s gravimetric water content (SWC) at 0–10 cm, 10–20 cm, and 20–30 cm soil layers. The SWC increased significantly in all three layers with the increase in watering amount [43,44].

### 4.4. Data Processing

The observed dates of seed release events were first transformed to day-of-year, and the duration of the seed release was calculated as the difference between the days of ESR and OSR. The relative change of the days (∆ days) was used to test the effects of water addition treatments on each event.
(1)$$∆\mathrm{days}=\frac{1}{n}\sum_{i=1}^{n}\left({day}_{treat}-{day}_{ctrl}\right)$$
where day_treat_ represents the day for a given event or the duration of the seed release in water addition plots, day_ctrl_ represents the corresponding day in control plots, and n represents the number of experimental years. The ∆ days < 0 showed that seed release events (duration of seed release) were advanced (shortened), while ∆ days > 0 meant that seed release events (duration of seed release) were delayed (prolonged) with water addition treatments.

### 4.5. Statistical Analysis

Simple linear regression analysis was used to determine the inter-annual trends and the relationships between the meteorological factors and seed release events. The Partial Eta Squared, η^2^, was used to represent the proportion of OSR variation explained by PFR and T_sum_. The effects of water addition treatments, year, and their interactions on seed release events over the seven years (2012–2018) were examined using linear mixed models. Water and year were used as fixed factors, while the plot was used as a random factor. The dependent factor was the timing of the onset and end of seed release (Type I Sum of Squares was used). Furthermore, Duncan post hoc tests were used to determine pairwise differences for significant effects. The effects of water addition treatments on the timing of different seed release events were evaluated separately for each year using one-way analysis of variance (ANOVA). The homogeneity of variances was tested by Levine’s tests, and one-sample Kolmogorov–Smirnov tests were used to validate the normality of the data’s distribution. All statistical analyses were completed in SPSS 20.0 (IBM, Inc.; Armonk, NY, USA), and Microsoft Excel 2019 (Redmond, WA, USA) was adopted for plotting.

## 5. Conclusions

The findings of the present study suggest that the seed release pattern of the desert shrub species *N. tangutorum* was significantly influenced by increased precipitation. Water addition treatments significantly accelerated the end of seed release. Moreover, the inter-annual variations of the onset and end of seed release were controlled by different regulators. The onset of seed release was jointly controlled by variation of the preceding fruiting phenology and mean summer temperature, while the end of seed release was controlled by cumulative summer precipitation. These results highlight the importance of water in regulating the phenology of desert plants besides temperature, and add to our previously published results [29,43,44]. However, further research on the influence of the size and fitness of the mother plants on seed release traits, seed dispersal and seedling establishment will help to better interpret the results of this study.

## Figures and Tables

**Figure 1 plants-12-00088-f001:**
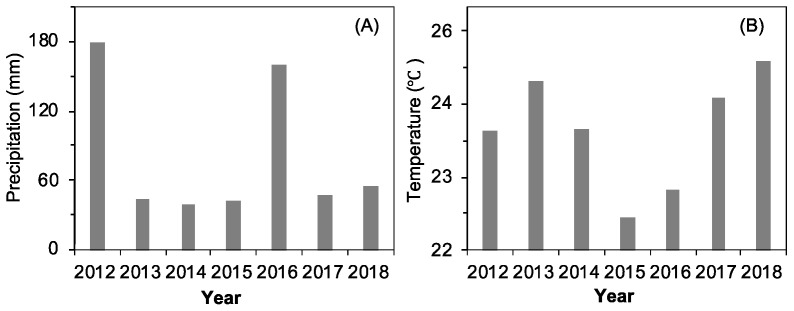
Variations in the accumulated precipitation (**A**) and average air temperature (**B**) from 1 June to 31 August of 2012–2018.

**Figure 2 plants-12-00088-f002:**
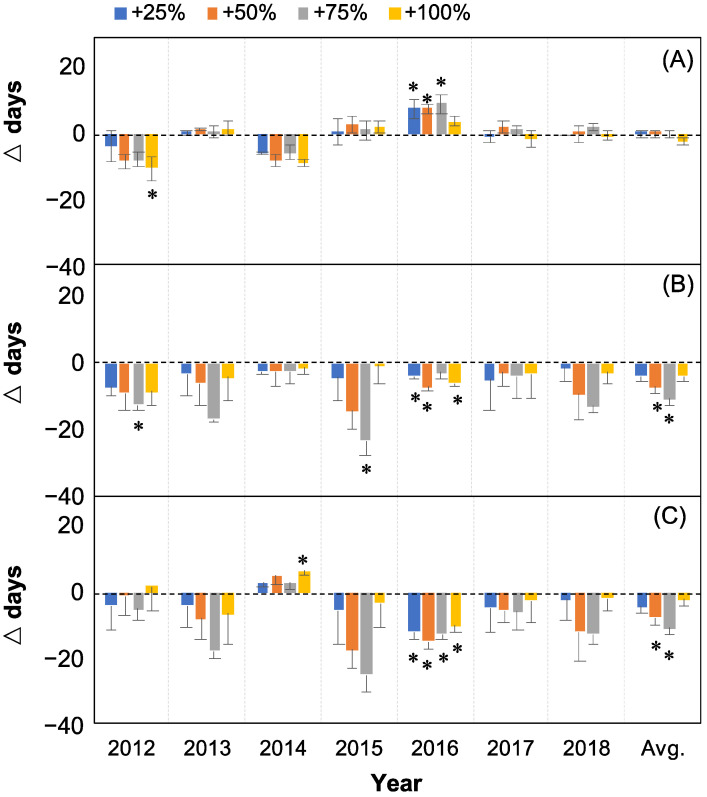
Relative changes compared to control (△days; mean ± SE) in the onset (**A**), end (**B**), and duration (**C**) of seed release in *Nitraria tangutorum* after the water addition treatments (+25%, +50%, +75%, and +100%) compared with the control. Positive and negative values represent relatively delayed or prolonged and advanced or shortened days, respectively. Avg. represents the average values from 2012 to 2018. The asterisk “*” marks indicate significant differences between the treatments compared with the control.

**Figure 3 plants-12-00088-f003:**
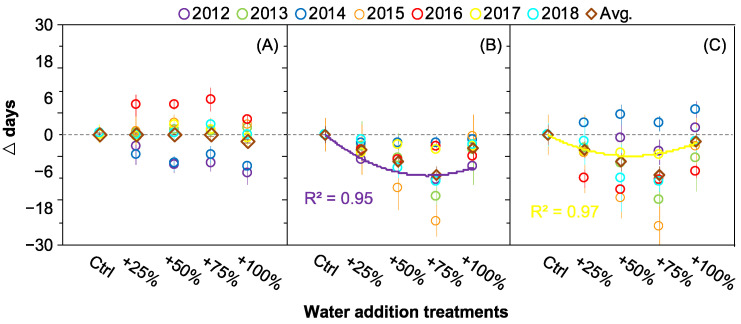
Correlations between the seed release onset (**A**), end (**B**), and duration (**C**) and the water addition treatments in *Nitraria tangutorum*. The solid lines indicate *p* ≤ 0.05.

**Figure 4 plants-12-00088-f004:**
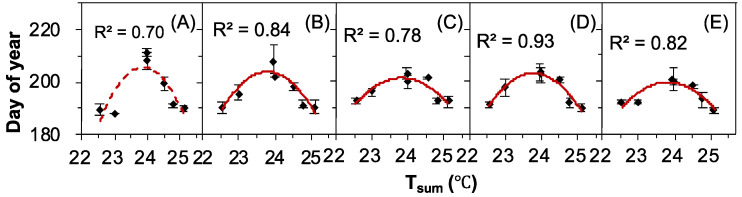
Correlations between the onset of seed release (OSR) and summer mean temperature (T_sum_) in *Nitraria tangutorum*. (**A**) Ctrl, (**B**) +25%, (**C**) +50%, (**D**) +75%, and (**E**) +100% water addition treatments. The solid lines indicate *p* ≤ 0.05, while the dashed line indicates 0.05 ≤ *p* ≤ 0.1.

**Figure 5 plants-12-00088-f005:**
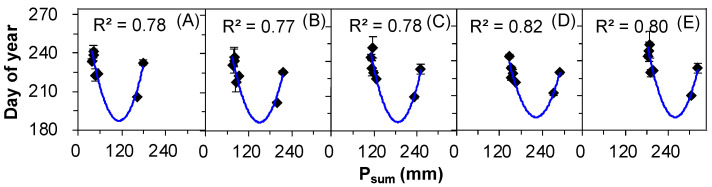
Correlations between the end of seed release (ESR) and summer cumulative precipitation (P_sum_) in *Nitraria tangutorum* under (**A**) Ctrl, (**B**) +25%, (**C**) +50%, (**D**) +75%, and (**E**) +100% water addition treatments. The solid lines indicate *p* ≤ 0.05.

**Figure 6 plants-12-00088-f006:**
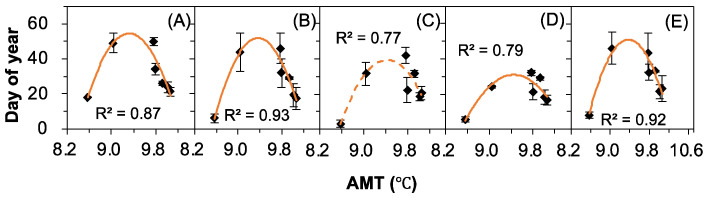
Correlations between the annual mean temperature (AMT) and the duration of seed release (DSR) of *Nitraria tangutorum* under (**A**) Ctrl, (**B**) +25%, (**C**) +50%, (**D**) +75%, and (**E**) +100% water addition treatments. The solid lines indicate *p* ≤ 0.05, while the dashed line indicates 0.05 ≤ *p* ≤ 0.1.

**Figure 7 plants-12-00088-f007:**
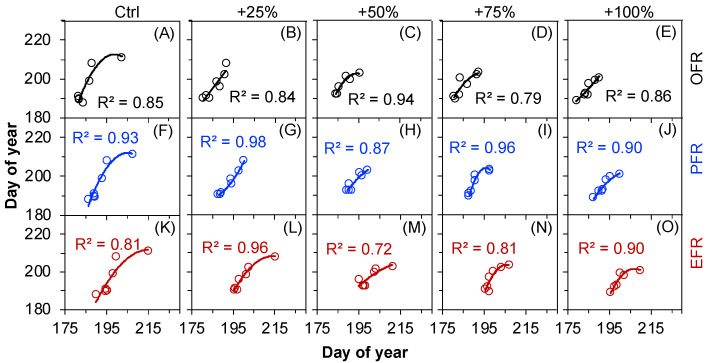
Correlations between the onset (**A**–**E**), peak (**F**–**J**), and end (**K**–**O**) of the fruit ripening events and the onset of seed release (OSR) of *Nitraria tangutorum*. The solid lines indicate *p* ≤ 0.05. OFR, PFR, and EFR represent the onset, peak, and end of the fruit ripening period, respectively.

**Figure 8 plants-12-00088-f008:**
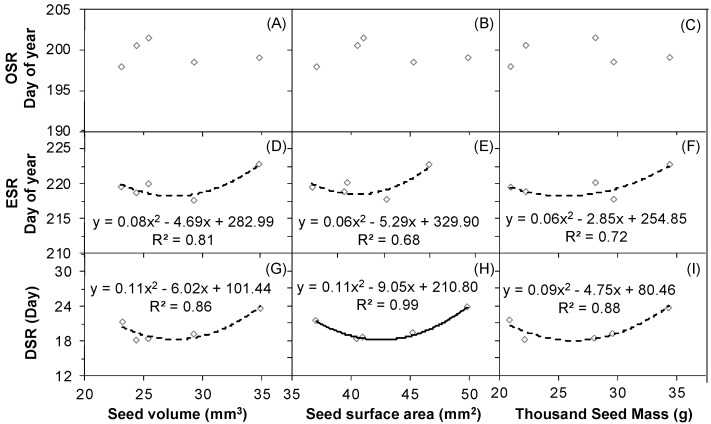
Correlations of the seed release (OSR), (ESR) and (DSR) with seed volume (**A**,**D**,**G**), seed surface area (**B**,**E**,**H**), and the Thousand Seed Mass (**C**,**F**,**I**) of *Nitraria tangutorum* in 2017. The solid line indicates *p* ≤ 0.05, and the dashed lines indicate *p* > 0.05. The seed volume, surface area, and the Thousand Seed Mass data were derived from a study by Zhang et al., 2019 [48].

**Table 1 plants-12-00088-t001:** F- and *p*-values of linear mixed model (MIXMOD) analysis of the fixed effects of water addition treatments (water), year, the preceding phenological event (peak of the fruit ripening, PFR), and their interactions with the onset (OSR), end (ESR) and duration of (DSF) seed release from 2012–2018.

Source	OSR		ESR		DSR	
	F	*p*	F	*p*	F	*p*
Water	2.85	0.04	2.85	0.04	7.43	0.00
Year	2.4	0.05	2.49	0.05	21.90	0.00
PFR	8.3	0.00	8.35	0.00	2.42	0.01
Water × Year	1.54	0.17	1.54	0.17	1.09	0.40
PFR × Year	2.72	0.02	2.72	0.02	1.23	0.31
Water × PFR	1.11	0.38	1.1	0.38	2.68	0.00
Water × Year × PFR	0.38	0.77	0.38	0.77	1.20	0.32

**Table 2 plants-12-00088-t002:** Correlations and *p*-values between the meteorological factors and seed release events.

Meteorological Factors	OSR									
	Ctrl		+25%		+50%		+75%		+100%	
	R^2^	*p*	R^2^	*p*	R^2^	*p*	R^2^	*p*	R^2^	*p*
AMP	0.15	0.46	0.36	0.41	0.18	0.67	0.21	0.63	0.19	0.69
P_Win_	0.08	0.92	0.13	0.68	0.32	0.24	0.37	0.37	0.20	0.42
P_Spr_	0.78	0.11	0.56	0.20	0.43	0.32	0.45	0.30	0.70	0.07
P_sum_	0.63	0.12	0.66	0.12	0.39	0.30	0.45	0.30	0.63	0.12
P_Aut_	0.13	0.71	0.27	0.53	0.32	0.47	0.32	0.46	0.13	0.81
AMT	0.65	0.09	0.71	0.08	0.80	0.04	0.74	0.07	0.60	0.15
T_win_	0.46	0.06	0.56	0.19	0.57	0.19	0.38	0.38	0.51	0.23
T_spr_	0.54	0.29	0.54	0.21	0.55	0.20	0.67	0.11	0.77	0.06
T_sum_	0.70	0.09	0.84	0.03	0.78	0.05	0.93	0.00	0.82	0.03
T_aut_	0.28	0.83	0.10	0.82	0.07	0.85	0.14	0.73	0.20	0.63
Meteorological Factors	ESR									
	Ctrl		+25%		+50%		+75%		+100%	
	R^2^	*p*	R^2^	*p*	R^2^	R^2^	*p*	R^2^	*p*	R^2^
AMP	0.49	0.37	0.44	0.58	0.73	0.23	0.73	0.39	0.41	0.47
P_Win_	0.19	0.62	0.18	0.50	0.03	0.83	0.06	0.77	0.13	0.59
P_Spr_	0.68	0.29	0.57	0.19	0.57	0.19	0.73	0.06	0.52	0.23
P_sum_	0.78	0.05	0.77	0.05	0.78	0.05	0.82	0.03	0.80	0.04
P_Aut_	0.50	0.14	0.48	0.27	0.38	0.39	0.04	0.75	0.53	0.22
AMT	0.69	0.15	0.77	0.06	0.62	0.15	0.54	0.18	0.74	0.07
T_win_	0.04	1.00	0.02	0.95	0.02	0.96	0.07	0.93	0.05	0.90
T_spr_	0.11	0.68	0.11	0.78	0.39	0.38	0.55	0.10	0.11	0.80
T_sum_	0.03	0.87	0.06	0.89	0.14	0.74	0.33	0.28	0.05	0.90
T_aut_	0.08	0.87	0.10	0.80	0.25	0.56	0.62	0.13	0.14	0.74
Meteorological Factors	DSR									
	Ctrl		+25%		+50%		+75%		+100%	
	R^2^	*p*	R^2^	*p*	R^2^	R^2^	*p*	R^2^	*p*	R^2^
AMP	0.09	0.83	0.13	0.76	0.45	0.30	0.45	0.30	0.14	0.74
P_Win_	0.36	0.41	0.26	0.45	0.12	0.58	0.04	0.74	0.25	0.44
P_Spr_	0.31	0.48	0.28	0.52	0.36	0.39	0.31	0.47	0.41	0.35
P_sum_	0.41	0.35	0.62	0.15	0.75	0.07	0.82	0.03	0.65	0.12
P_Aut_	0.79	0.05	0.63	0.14	0.47	0.28	0.27	0.53	0.64	0.13
AMT	0.87	0.02	0.93	0.01	0.77	0.06	0.79	0.05	0.92	0.03
T_win_	0.17	0.69	0.16	0.70	0.06	0.88	0.12	0.77	0.11	0.79
T_spr_	0.12	0.78	0.12	0.77	0.17	0.68	0.26	0.55	0.08	0.85
T_sum_	0.44	0.31	0.32	0.46	0.11	0.80	0.16	0.70	0.23	0.59
T_aut_	0.06	0.89	0.03	0.94	0.15	0.71	0.30	0.49	0.04	0.92

## Data Availability

All data are presented in the main text.

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
