# Peer review of "Contrasting Regulators of the Onset and End of the Seed Release Phenology of a Temperate Desert Shrub Nitraria tangutorum"

_plants, 2022, doi:10.3390/plants12010088_

Round 1
Reviewer 1 Report
The presented results look very interesting. Still, I have some remarks on the current manuscript:
General remarks:
- In the statistical analysis, I miss the putative influence of the size and fitness of the mother plants on the seed release traits. These kind of variables could be added in the fixed part of the mixed models to check whether they also influence the response variables. I also wonder if the size and weight of the seeds themselves may influence their release. Also these type of variables could have been (or should have been) added in the mixed models.
- In fact, much or all of the correlation studies could be omitted by adding the variables in the fixed part of the models.
- English language is poor. Sometimes I doubted how to interpret the sentences. Sometimes verbs seem lacking, sometimes connecting words seem to lack. Also typing mistakes are present. And sometimes I cannot figure out what is the meaning of the sentences. Sometimes Latin plant names are not italic….
- Introduction is too short. The topic is not properly presented, nor the context. I find interesting information in the discussion that should fit in the introduction.
Some detailed remarks:
- Line 56: leaf cessation: is this leaf senescence or leaf fall?
- Lines 60 – 63: what is said here? Rephrase and also explain better.
- Lines 66 – 69: explain better
- Line 70 – 71: This is a fundamental part of the introduction. Only 1 reference? And from a long time ago (1994)? This should be worked out in more detail, including more recent findings in this field. Compare with non-desert species, what about other traits than phenology,… (describe the context of your research). Give a broader view on the topic.
- Line 90: in which time frame? How many years were looked at?
- Figure 1: make two plots
- Lines 103 – 109: When you have a significant interaction term in a model, it is of not much use to look at the significance of the individual variables, as the significance of the first depends on the second and vice versa.
- Table 1: you should show more than only p-values. A significant effect can both have a strong or a small influence. This can be deduced from other test statistics.
- Line 117: also show and refer to the test statistics
- Line 118: amounts of what?
- Figure 2: why not showing the modeled data rather than the raw data (put raw data in a supplementary material)?
- Line 129-130: This may be an interesting observation, but I do not find any hypothesis or description of it in the discussion?
- Line 132 … do not forget to refer to Table 2. Also add correlation coefficients in the table, not only the p-values
- First paragraph of 2.5: there is a mention of leaf phenological events, and also of fruit ripening events. It was not clear to me what these fruit ripening events were.
- Line 203-205: explain better the results of the mentioned earlier study. Also explain why they are “similar” to these results. Explain, or hypothesise why the results are similar.
- Line 207-208: you mention smaller seeds. This is the reason why seed size and seed weight might play a role in the seed release, and should be added in the mixed models.
- Figure 8 seems not really necessary here. Describe properly in the text, can the data be used in the mixed models of this paper?
- Line 216 – 227: this paragraph seems not in its place here. You did not do any research on rodents or ants. This can be integrated in the introduction.
- Line 228… : same remark: why the seed size and seed weight was not added in the mixed models?
- Line 241: it is not clear to me what is your “opinion”. Explain better
- Line 256 – 276: this information seems to fit more for the introduction than for the discussion
- Paragraph 4.5: show the formulas of the mixed models
- Line 339: “influenced”: describe in which way the influence works
- Line 342: was plant (I guess seedling fitness is meant) fitness measured in this research?
- Line 342 – 345: this sentence is already present in the discussion, is a mere repetition here. As far as I can see, the seedling adaptation or fitness was not studied here.
Author Response
General remarks:
- In the statistical analysis, I miss the putative influence of the size and fitness of the mother plants on the seed release traits. These kind of variables could be added in the fixed part of the mixed models to check whether they also influence the response variables. I also wonder if the size and weight of the seeds themselves may influence their release. Also these type of variables could have been (or should have been) added in the mixed models.
Response:
We really appreciate your great comments on our manuscript. These comments are very helpful in improving the quality of the paper.
Plots/Nebkhas with shrubs that in similar size and fitness were selected when we set our Water Addition Platform. Water addition treatment did have accelerated the growth of Nitraria tanturorum in our study site. Unfortunately, these observations were not conducted every year but in one or two years. These short term data are not suitable to be added in our mixed model in the present manuscript. However, we correlated them with the seed release phenology in the observation year instead. 2.6 The correlations between seed release events and seed size and weight with a brand new Figure (Figure 8) that shown how the seed size and weight influence the seed release in 2017 has been added in as a independent paragraph and the results have been discussed in 3.1 paragraph 2 in the revised manuscript.
As for the putative influence of the size and fitness of the mother plants on the seed release, we correlated the AGB with the seed release time instead. The results showed that the relationships between seed release and AGB were opposite between the two observation years (Figure 1) and it is hard to make a clear conclusion. Considering the large inter-annual variability in phenology events and the representativeness of the results of the two years, to be caution, we choose keeping the result in our own plate temporally. But, your words, influence of the size and fitness of the mother plants on the seed release traits, has been cited and fixed in the 5. Conclusions in the revised manuscript to inspires readers to think of it. We will defenitely design an experiment to answer the question in the following years.
Figure 1 Correlations between OSR(A, D), ESR(B, E), DSR(C, F) and AGB in 2013 and 2014. The dashed lines indicate P > 0.05 and the solid lines indicate P ≤ 0.05.
- In fact, much or all of the correlation studies could be omitted by adding the variables in the fixed part of the models.
Response:
Thanks for your comment. One factor, PFR, has been added in the mixed linear model in the revised manuscript. Which brought changes to the results of the linear mixed model. The correlation figures were kept to show the non-linear relationships clearly to readers.
- English language is poor. Sometimes I doubted how to interpret the sentences. Sometimes verbs seem lacking, sometimes connecting words seem to lack. Also typing mistakes are present. And sometimes I cannot figure out what is the meaning of the sentences. Sometimes Latin plant names are not italic….
Response:
I apologize for the trouble brought to you by my poor English writing. So, the revised manuscript had been sent out for language editing by a native English speaker from MogoEdit company.
- Introduction is too short. The topic is not properly presented, nor the context. I find interesting information in the discussion that should fit in the introduction.
Response:
The introduction has been reconstructed and rewritten according to your comments. Please see details in the revised manuscript.
Some detailed remarks:
- Line 56: leaf cessation: is this leaf senescence or leaf fall?
Response:
Leaf cessation including both leaf senescence and leaf fall in out study. The confusion has been fixed in the revised manuscript.
- Lines 60 – 63: what is said here? Rephrase and also explain better.
Response:
These sentences have been rewritten, please see details in the revised manuscript.
- Lines 66 – 69: explain better
Response:
These sentences have been rewritten, please see details in the revised manuscript.
- Line 70 – 71: This is a fundamental part of the introduction. Only 1 reference? And from a long time ago (1994)? This should be worked out in more detail, including more recent findings in this field. Compare with non-desert species, what about other traits than phenology,… (describe the context of your research). Give a broader view on the topic.
Response:
Again, thanks for your good comments here. The introduction has been reconstructed and rewritten according to your comments. Please see details in the revised manuscript.
- Line 90: in which time frame? How many years were looked at?
Response:
The ambiguity has been corrected, please see details in the revised manuscript.
- Figure 1: make two plots
Response:
The figure has been reploted, please see details in the revised manuscript.
- Lines 103 – 109: When you have a significant interaction term in a model, it is of not much use to look at the significance of the individual variables, as the significance of the first depends on the second and vice versa.
Response:
Thanks for your statistical lesson. I do have learned from you about this.
- Table 1: you should show more than only p-values. A significant effect can both have a strong or a small influence. This can be deduced from other test statistics.
Response:
Table 1 has been fixed, please see details in the revised manuscript.
- Line 117: also show and refer to the test statistics
Response:
This has been fixed, please see details in the revised manuscript.
- Line 118: amounts of what?
Response:
This has been fixed, please see details in the revised manuscript.
- Figure 2: why not showing the modeled data rather than the raw data (put raw data in a supplementary material)?
Response:
Thanks for your comments. The figure was used to show the inter-annual variation clearly and to compare the inter-annual consistency between the onset and end of the seed release. As a result, this figure was kept in the revised manuscript.
- Line 129-130: This may be an interesting observation, but I do not find any hypothesis or description of it in the discussion?
Response:
Thanks for your comments.This has been added in the discusstion in the revised manuscript.
- Line 132 … do not forget to refer to Table 2. Also add correlation coefficients in the table, not only the p-values
Response:
This has been fixed. Table 2 has been redone. Please see details in the revised manuscript.
- First paragraph of 2.5: there is a mention of leaf phenological events, and also of fruit ripening events. It was not clear to me what these fruit ripening events were.
Response:
This has been clearified both in this location and in Method. Please see details in the revised manuscript.
- Line 203-205: explain better the results of the mentioned earlier study. Also explain why they are “similar” to these results. Explain, or hypothesise why the results are similar.
Response:
These sentences have been reconstruction and rewritten. Please see details in the revised manuscript.
- Line 207-208: you mention smaller seeds. This is the reason why seed size and seed weight might play a role in the seed release, and should be added in the mixed models.
Response:
Here, we have exactly the same response to the first one. Please see details on the top.
- Figure 8 seems not really necessary here. Describe properly in the text, can the data be used in the mixed models of this paper?
Response:
It has been removed. Thanks for your suggestion. Please see details in the revised manuscript.
- Line 216 – 227: this paragraph seems not in its place here. You did not do any research on rodents or ants. This can be integrated in the introduction.
Response:
According to your comments, this paragraph has been integrated in the introduciton. Please see details in the revised manuscript.
- Line 228… : same remark: why the seed size and seed weight was not added in the mixed models?
Response:
Here, we have exactly the same response to the first one. Please see details on the top.
- Line 241: it is not clear to me what is your “opinion”. Explain better
Response:
These sentences have been rewritten. Please see details in the revised manuscript.
- Line 256 – 276: this information seems to fit more for the introduction than for the discussion
Response:
According to your comments, this sentences has been integrated in the introduciton. Please see details in the revised manuscript.
- Line 339: “influenced”: describe in which way the influence works
Response:
This paragraph has been rewritten. Please see details in the revised manuscript.
- Line 342: was plant (I guess seedling fitness is meant) fitness measured in this research?
Response:
Your are right. This has been fixed in the revised manuscript.
- Line 342 – 345: this sentence is already present in the discussion, is a mere repetition here. As far as I can see, the seedling adaptation or fitness was not studied here.
Response:
Thanks for your comments again. This paragraph has been reconstructed and rewriten. Please see details in the revised manuscript.

Reviewer 2 Report
There are a number of minor typological errors throughout the document that need to be addressed by the authors. I have tabulated these in the document attached.
My major concern with the paper is its proximity to
Preceding Phenological Events Rather than Climate Drive the Variations in Fruiting Phenology in the Desert Shrub Nitraria tangutorum
with which it seems to share much of the data, analysis and figures.
This article is perhaps the fifth of a series discussing the same phenomena and/or using the data collected from the same experiments.
In the conclusions to the above paper the authors stated
'Mores studies across species are needed in the future to test the universality of the present results in a desert ecosystem.'
They should in my opinion clarify how this paper is different from the previously published one.

Author Response
Point 1. There are a number of minor typological errors throughout the document that need to be addressed by the authors. I have tabulated these in the document attached.
Response:
I apologize for the trouble brought to you by my poor English writing.
Thanks for your editing.
All your tabulted errors have been fixed in the revised manuscript which had been sent out for language editing by a native English speaker from MogoEdit company. The language editing certification was also attached together with the revised manuscript.
Point 2. My major concern with the paper is its proximity to Preceding Phenological Events Rather than Climate Drive the Variations in Fruiting Phenology in the Desert Shrub Nitraria tangutorum with which it seems to share much of the data, analysis and figures.
Response:
I can understand your concern. The whole series of the phenological events including the onset, peak, and end of leaf unfolding, flowering, fruiting, seed release, leaf coloring, leaf fall, braches elongation of N. tangutorum have been recored in our Water Additon Platform.
Preceding Phenological Events Rather than Climate Drive the Variations in Fruiting Phenology in the Desert Shrub Nitraria tangutorum is our last published paper which reported the effects of water addition treatment on the fruiting phenology. The present manuscript will report the influence of water addition on the seed release phenology. The two paper did share similar statistical analysis.
Point 3. This article is perhaps the fifth of a series discussing the same phenomena and/or using the data collected from the same experiments.
Response:
Yes, this manuscript will be our fouth paper of the series. Our next paper will focus on the influence of water addition treatment on the flowering phenology.
Point 4. In the conclusions to the above paper the authors stated 'Mores studies across species are needed in the future to test the universality of the present results in a desert ecosystem.' They should in my opinion clarify how this paper is different from the previously published one.
Response:
Thanks very much for your comments. According to your comments, the difference and the cotinuation of this paper from the published ones have been interpreted in the Conclusion of the revised manuscript.

Round 2
Reviewer 1 Report
The quality of the manuscript has improved a lot!
Reviewer 2 Report
I am pleased that the revisions have all been incorporated.